# Agreement between an Image-Based Dietary Assessment Method and a Written Food Diary among Adolescents with Type 1 Diabetes

**DOI:** 10.3390/nu13041319

**Published:** 2021-04-16

**Authors:** Laura Heikkilä, Marja Vanhala, Raija Korpelainen, Päivi Tossavainen

**Affiliations:** 1Department of Sports and Exercise Medicine, Oulu Deaconess Institute Foundation sr., FI-90101 Oulu, Finland; marja.vanhala@odl.fi (M.V.); raija.korpelainen@odl.fi (R.K.); 2Center for Life Course Health Research, Faculty of Medicine, University of Oulu, FI-90014 Oulu, Finland; 3Medical Research Center, Oulu University Hospital and University of Oulu, FI-90014 Oulu, Finland; paivi.tossavainen@oulu.fi; 4Department of Pediatrics and Adolescents, Oulu University Hospital, FI-90029 Oulu, Finland; 5PEDEGO Research Unit, University of Oulu, FI-90014 Oulu, Finland

**Keywords:** feasibility studies, smartphone, energy intake, carbohydrates, type 1 diabetes, youth

## Abstract

Valid and useful dietary assessment methods for adolescents with type 1 diabetes (T1D) are needed. In this study, we compared an image-based method with a written food diary for dietary intake estimation among adolescents with T1D and evaluated the adolescents’ experiences of the methods. Adolescents with T1D aged 13 to 18 years (*n* = 13) photographed their meals (*n* = 264) with a mobile phone camera and simultaneously kept a written food diary for four consecutive days. The participants filled out electronic background and feedback questionnaires. The agreement between the methods was evaluated using intraclass correlation coefficients (ICCs) and Bland–Altman plot analyses. The agreement between the methods was moderate to excellent for the energy intake (ICC = 0.91, 95% confidence interval (CI): 0.66 to 0.97, *p* < 0.001) and good to excellent for total carbohydrate intake (ICC = 0.95, 95% CI: 0.84 to 0.99, *p* < 0.001). The adolescents considered photographing easier and faster than keeping a food diary. In conclusion, the image-based method appeared comparable to the food diary for dietary intake estimation among adolescents with T1D. The photographing of meals may become a useful dietary assessment tool for adolescents with T1D, but must be further developed and validated.

## 1. Introduction

The reliable measurement of dietary intake is necessary in nutrition research. Accurate dietary intake data are especially necessary in the treatment of type 1 diabetes (T1D), where nutrition counselling supports healthy eating habits and is an important part of diabetes self-treatment and good metabolic control [1]. Traditional dietary assessment methods, such as a food diary and a food frequency questionnaire, are useful but have limitations [2]. These methods are time-consuming and burdensome for patients and can result in reporting errors and deviations from a normal diet [3].

Previous studies have underlined challenges in the measurement of dietary intake in adolescents, such as the tendency to misreport [4,5,6,7,8]. In particular, underreporting due to missing foods and difficulties in estimating portion sizes have been reported [8,9]. Food records were found to underestimate energy intake by 19–41% compared with the doubly labelled water method in children and adolescents [10]. Underreporting is more common among girls than in boys and in obese compared to non-obese adolescents [4,11]. Worries about self-image and rebellion against authorities are also typical at this stage of life and can result in poor adherence to study protocols [11].

Adolescents with T1D have to count meal carbohydrate amounts several times a day to match insulin doses and to manage varying blood glucose levels. Thus, feasible dietary assessment methods are particularly important among young people with diabetes. Using technology, such as smartphones and image-based methods, in dietary assessment could reduce the burden on adolescents and improve compliance with reporting their dietary habits [12]. Some image- or voice-based carbohydrate intake estimation methods have been developed among adults with diabetes; however, further development and validation of these are required [13,14].

In image-based dietary assessment studies, in general, the participants are asked to photograph their meals before eating and to send the images with further information to a researcher. Thereafter, the image data are analyzed to specify the foods, to estimate portion sizes, and to count the dietary intake. Recent studies suggested that image-based methods could provide more accurate dietary data compared with traditional methods [15]; however, their superiority has not been demonstrated [16]. Image-based dietary records have been found to be feasible and valid dietary assessment methods and comparable to the traditional methods in adults [16,17,18,19,20,21].

Among children and adolescents, image-based dietary assessment methods also appear to have similar validity compared to traditional methods [22,23,24,25,26]. In some studies, children and adolescents found image-based methods easier and faster than the traditional methods [22,23,24,27]. Although the importance of dietary intake assessment in diabetes is well known, the validity of image-based methods has not been studied among adolescents with T1D to our knowledge. In this study, we aimed to estimate the agreement between an image-based method and a written food diary in adolescents with T1D. In addition, adolescents’ experiences of the usability of the methods were evaluated.

## 2. Materials and Methods

### 2.1. Participants and Study Design

The study population comprised sixteen adolescents with T1D who were aged 13 to 18 years. The participants were recruited from the Pediatric Diabetes Outpatient Clinic at Oulu University Hospital (Oulu, Finland). The study was conducted according to the guidelines laid down in the Declaration of Helsinki, and all procedures involving human subjects were approved by the Northern Ostrobothnia District Hospital Ethics Committee, Oulu, Finland (record number 46/2015; accepted 20 April 2015). Written informed consent was obtained from all the participants and also from their parents or guardians if the child was under 15 years old.

In this comparative study, the image-based method was evaluated and compared with the written food diary as a standard method. The participants received written and oral instructions to take photographs of meals at a 45-degree angle before and after eating and to concurrently maintain a food diary over four consecutive days. The subjects were asked to make sure that all ingredients were clearly seen in the photograph and to photograph second servings and snacks too.

The participants were asked to judge the clarity of photographs they had taken and to retake photographs that were unclear. Paying attention to the lighting and using the camera flash, if necessary, were advised. If a meal was not eaten completely, the leftovers also had to be photographed. The ingredients of mixed foods were asked to be provided in detail (e.g., sandwich: sliced bread, margarine, ham, lettuce, and cucumber). In the written food diaries, the participants were instructed to estimate portion sizes as pieces (e.g., two small apples), household measures (e.g., dL), size (e.g., 2 × 5 × 5 cm), or weight (grams).

The participants sent the photographs to a researcher using the WhatsApp application (WhatsApp Inc., Menlo Park, CA, USA) the day after photographing. Further details were sent as a text or an audio message. The participants provided additional details regarding the food quality (e.g., fat-free or semi-skimmed milk, and brand of packaged food) and cooking methods. The researcher verified the pictures after the four-day study period, sent a feedback message, and asked for additional information from the participants if necessary. The participants returned the written food diaries by mail at the end of the study period. Additional questions about the quality and amount of food in the written food diaries were not asked.

The adolescents replied to electronic background and feedback questionnaires (Surveypal Oy, Tampere, Finland). The background questionnaire included information on the date of birth, gender, date of diabetes onset, latest hemoglobin A1c (HbA1c; measured less than three months ago), and use of a mobile phone’s camera. The actual height and weight measurements were drawn from the medical records. The participant experiences of the usability of the image-based method and sending photographs and further dietary information were evaluated with the feedback questionnaire.

The ease and quickness of the methods were estimated using four questions with a 5-point Likert scale (1 = very easy to 5 = very difficult or 1 = very quick to 5 = very slow). The factors restricting the use of the image-based method were asked using a multiple-choice question: “Did you have any problems with using the image-based method?” The answer options were as follows: (1) I forgot to take photos; (2) I did not carry my mobile phone with me; (3) my mobile phone’s battery was out of charge; (4) my mobile phone’s camera was poor; (5) lightning was poor; (6) mobile phone use was not allowed; (7) I was too embarrassed to take photos; (8) something else, what?; and (9) I had no problems. The enabling factors were queried using an open-ended question: “Which factors made the image-based method easy?” In addition, the adolescents were asked which methods they would prefer in the future.

The same nutritionist (L.H.) analyzed both the photographs and food diaries using the AivoDiet software program, version 2.0.2.3 (Aivo Finland Oy, Turku, Finland). The photographs were analyzed first, and the food diaries at least four weeks later. The foods in the photographs were identified visually and using further detailed information provided by the participants.

Two picture booklets of the portion sizes were employed in estimating the food amounts in the photographs [28,29,30,31]. The booklets contained pictures and weights for most typically consumed foods and ingredients in Finland in three to four portion sizes. The portion sizes of photographed food and drinks from the participants were compared to standard portion sizes and entered into the AivoDiet^®^ software as grams. The AivoDiet^®^ software uses the Finnish food composition database Fineli (National Institute for Health and Welfare, Helsinki, Finland), which includes the average energy and nutrient contents of foods and food products used in Finland.

The following assumptions were made: (1) if exactly the same picture was received in duplicate, it was considered to depict the same meal; (2) the absence of a leftover photograph indicated the meal was finished; (3) if the meal was not photographed, but its components were given in further detail as a text or audio message, it was included in the analysis; (4) if the ingredients of a mixed dish were not provided in further detail, it was assumed to have been cooked with a conventional Finnish recipe; and (5) if a portion size was not provided in the food diary, standard portion sizes were used.

### 2.2. Statistical Analyses

The mean daily energy and nutrient intakes were calculated from the photographs and from the food diaries. First, the analyses were made using the original data. Second, to evaluate the image-based method’s validity in energy and nutrient intake estimation, the meals that the adolescents forgot to photograph were deducted from the food diary data (later, deducted data). The normality of the continuous variables was tested. The paired samples *t*-test was used to estimate the differences between the methods in the energy and nutrient intake estimation when the difference between the two variables was normally distributed. The Wilcoxon signed rank test was employed when the difference between the variables was non-normally distributed.

Agreement between the methods was estimated using Bland–Altman plots and intraclass correlation coefficients (ICCs). Differences between the methods on the Y-axis were plotted against the mean of the two methods on the X-axis. The limits of agreement (LOA; mean difference ± 2 standard deviation units) were marked on the Bland–Altman plots. The presence of a proportional bias (i.e., increase/decrease in the variability of the differences as the magnitude of the measurements increased) was tested using linear regression analysis.

The ICC estimates and their 95% confidence intervals (CIs) were analyzed based on a single-rating, absolute-agreement, two-way mixed-effects model. For non-normally distributed variables, the ICCs were calculated using logarithmic values. ICCs less than 0.50 were interpreted as poor, 0.50 to 0.75 as moderate, 0.75 to 0.90 as good, and greater than 0.90 as excellent agreement [32]. A *p*-value <0.05 was considered statistically significant. The statistical analyses were performed using IBM SPSS Statistics, version 23 (IBM Corporation, Armonk, NY, USA).

## 3. Results

Sixteen adolescents with T1D aged 13 to 18 years participated, and thirteen of them were included in this study (Table 1). Three female subjects were excluded due to missing food diaries. Nine (69%) of the thirteen participants were females. The diabetes duration of the participants ranged from seven months to fourteen years. Eight (62%) of the subjects had normal weight, two (15%) were underweight, and three (23%) were overweight, defined by weight-for-height [33].

The participants photographed on average 18 ± 2 and recorded 20 ± 2 meals in their food diaries during the four-day study period. Two participants forgot to photograph one meal and three participants forgot more than one meal (range 7 to 10). Altogether, 237 meals from the photographs and 264 meals from the food diaries were analyzed in this study.

### 3.1. Energy and Nutrient Intake Estimates

In the original data, the image-based method underestimated the median energy intake by 8.5% compared with the food diary (median difference −153 kcal/day, interquartile range (IQR) 251, *p* = 0.010). The difference in the median energy intake between the image-based method and the food diary was 34.4% (−804 kcal/day, IQR 1347) in boys and 7.2% (−113 kcal/day, IQR 215) in girls. Table 2 represents the energy and nutrient intakes estimated from the photographs and the food diaries in the original data.

In the deducted data, the agreement between the methods was moderate to excellent in the energy intake estimation (ICC = 0.91, 95% CI: 0.66 to 0.97, *p* < 0.001) (Table 3). The image-based method underestimated the mean energy intake by 5.5% (mean difference −88 ± 131 kcal/day, *p* = 0.033). Underestimation of the mean energy intake was 9.8% (−167 ± 85 kcal) in boys and 3.4% (−53 ± 136 kcal) in girls. The ICCs between the image-based method and the food diary were at least moderate for most nutrients in the deducted data (range = 0.61–0.99, *p* < 0.01). Regarding the carbohydrate intake estimation, agreement between the methods was good to excellent (ICC = 0.95, 95% CI: 0.84 to 0.99, *p* < 0.001). The mean carbohydrate intake did not differ between the methods (−8.4 ± 17.7 g/day, *p* = 0.114).

Agreement between the methods was poor to excellent in the fat (ICC = 0.80, 95% CI: 0.48 to 0.94, *p* < 0.001) and protein intake estimation (ICC = 0.72, 95% CI: 0.24 to 0.91, *p* < 0.001). The daily fat intake estimates did not differ between the methods (−3.5 ± 8.2 g/day, *p* = 0.148). The image−based method underestimated the mean protein intake (−5.1 ± 7.2 g/day, *p* = 0.025).

The Bland–Altman plots for energy (mean difference −88 kcal, 95% CI: −167 to −8, LOA: −350 to 175) and carbohydrate (−8.4 g, 95% CI: −19.0 to 2.3, LOA: −43.7 to 27.0) intakes showed slight positive trends, indicating that the image-based method tended to underestimate energy and carbohydrate intakes when the mean intakes were low (Figure 1a,b). Similar trends were not found in the protein (−5.1 g, 95% CI: −9.4 to −0.8, LOA: −19.5 to 9.3) or fat (−3.5 g, 95% CI: −8.5 to 1.4, LOA: −20.0 to 13.0) intake estimation (Figure 1c,d). In linear regression models, there was no proportional bias between the methods in the energy, carbohydrate, protein, and fat intake estimation.

### 3.2. Experiences of the Image-Based Method

According to the feedback questionnaire, 11 of the 13 subjects (85%) considered recording food intake by a mobile phone very or quite easy, and 12 (92%) assessed the method as very or quite quick. All the participants (100%) experienced sending photographs as being very or quite easy, and 11 subjects (85%) considered sending further information to be very or quite easy.

The reported reasons for the ease of the use of the image-based method were reported as (1) I think the method was quick to use (46%); (2) I always carry my mobile phone with me (38%); (3) I am used to taking photos with a mobile phone camera (23%); and (4) I received clear instructions to take photographs (15%). The two most common reported problems with the image-based method were forgetting to take photos (62%) and poor lightning (31%).

When the participants were asked which of the methods they would prefer if they were to record their food intake in the future, seven (54%) participants chose the image-based method, five (39%) participants accepted both methods, and one (8%) participant preferred the food diary. The reported reasons for preferring the image-based method were the easiness and quickness.

## 4. Discussion

In the present study, the image-based method appeared to be comparable to the written food diary regarding the energy and nutrient intake estimation of adolescents with T1D. The image-based method slightly underestimated the mean energy intake, while agreement between the methods was good in the carbohydrate intake estimation. We found that the image-based method was a feasible method among adolescents with T1D; however, young people’s tendency to forget to take photos weakened its accuracy. Furthermore, the adolescent participants preferred the image-based method in dietary assessment according to the feedback questionnaire. Although there was a relatively low number of participants in the present survey, the adolescent participants recorded a total of 264 meals, snacks, and drinks during the study period.

The present study showed that the image-based method had moderate to excellent agreement with the food diary in energy intake estimation and good to excellent agreement in carbohydrate intake estimation among adolescents with T1D. Our results are in accordance with previous studies where the intakes of energy and carbohydrate were compared using image-based methods and a written food diary among children and adolescents [24,34].

In the present study, the mean difference in the carbohydrate intake between the methods was 8 g per day (4.4%); thus, the methods can be considered similar in practice. The mean energy intake was 88 kcal (5.5%) per day lower when assessed using the image-based method compared with keeping a food diary. This systematic bias was equal to earlier studies that reported 53–222 kcal per day underestimation of energy intake among children and adolescents [24,25,34].

The ICCs between the two dietary assessment methods were at least moderate for most nutrients, reflecting that the methods were comparable in estimating the adolescents’ dietary intake. The image-based method was equal to the food diary in fat intake estimation but underestimated the protein intake. However, it can be debated whether the mean underestimation of protein intake in this study (5 g/day) would have clinical or practical relevance.

The protein intake estimates had weaker correlations than those of other macronutrients in this study. Davison et al. [24] and Wang et al. [34] also found weaker correlations in protein intake estimation compared with those of other macronutrients among children and adolescents. Similarly, the mean underestimation of protein intake was low (3 to 7 g/day) in these two previous studies [24,34].

In this study, we removed the foods that adolescents forgot to photograph from the food diary data to evaluate more accurately the image-based method’s ability to estimate dietary intake. According to the original data, the image-based method underestimated the median daily energy intake by 153 kcal (8.5%). The underestimation in our results was comparable with those of some previous studies among children and adolescents [24,34]. Adults in earlier reports were found to underreport their energy intake by approximately 15% [35].

Underestimation of the mean carbohydrate intake in the original data (34.3 g/day, 15.7%) was greater when compared with one recent study [34]. Food content estimation in diabetes is very much carbohydrate-centered. Adolescents with T1D in this study might have reported carbohydrate-rich foods’ portion sizes more accurately in food diaries compared to children and adolescents in general.

Based on the present and previous studies, commitment to studies might be challenging for adolescents. However, people with T1D are generally used to following instructions to maintain good glucose control, which can also improve their ability to follow protocols. In the present study, forgetting to take photographs was one source of misreporting, and we found that this weakened the feasibility of the method. Forgetting to photograph has also been found to be a source of underreporting in previous surveys [22,36].

The typical forgotten foods in the present study were drinks and extra snacks, for example, chips or meatballs. Systematic misreporting also existed: for example, a bread spread was not always seen in the photographs. A reduction of missing photographs could possibly be reached with prompts and systematic contact with the subjects during the survey. Backup methods, more detailed instructions, and final inspections of food diaries might also be helpful to diminish missing data in future studies.

According to the present study, the image-based method appeared to be more accurate among the girls than in the boys. One explanation might be that adolescent girls were more interested in recording their dietary intake than boys, and therefore had a higher compliance with using the image-based method. In previous studies, underreporting of dietary intake has been more common among girls than in boys [11]. If the girls have also underreported their dietary intake in the written food diaries, it might have led to more similar estimates between the methods in this study. However, the small sample size might have resulted in selection bias and the observed gender differences.

One strength of the image-based method is that it exerts a lower respondent burden, as adolescents do not have to specify all consumed food and drinks and estimate portion sizes. On the other hand, the responsibility for portion size estimation is transferred to the researchers, which increases their burden and requires well-trained professionals for consistent and accurate imaged-based dietary assessment. Regular photograph inspection and contact with the participants also increases the researchers’ work. In this study, the researcher estimated portion sizes visually and using picture booklets of the portion sizes. Additionally, package labels and dishes seen in the photographs were of use. A fiducial marker was not used in this study but would be recommended for future studies to ensure the accuracy of portion size estimation.

The image-based method appears to be a feasible tool in dietary intake estimation. The development and improvement of feasible dietary assessment methods could be very important for people with T1D as they have to count the carbohydrate contents of their meals several times a day to adjust insulin doses and to estimate blood glucose changes. Advanced methods for intensive insulin delivery and continuous glucose monitoring by sensor devices are already available.

Novel automatic carbohydrate counting methods have already been developed; however, further research is needed to bring them into clinical practice [13,14]. Owing to the image-based method’s easiness and quickness, it could also benefit nutrition counselling among adolescents with T1D in general. To make the use of the image-based method easier and to diminish missing data, developing mobile applications for dietary data collection would be useful.

To our knowledge, the present study is the first image-based method validation study in adolescents with T1D. The fact that participants with T1D in this study were used to paying attention to their dietary habits and, thus, were experienced in estimating their food content and portion sizes might limit the generalizability of the results for healthy adolescents. In this study, we compared the image-based method with a food diary, which has limitations, such as being prone to reporting errors [35].

The written food diaries were returned by mail, and additional questions regarding the quality and amount of food were not asked, which may have weakened the validity of the diaries. In addition, if participants had sent photographs of the filled out written food diary to the researcher, the loss of some data may have been avoided. Only one rater analyzed the photographs, and a fiducial marker was not used, which may have limited the accuracy of the portion size estimation in this study. The feasibility and validity of the image-based method should be confirmed in larger studies.

## 5. Conclusions

The image-based method seemed to have good agreement with the written food diary for estimating the dietary intake of adolescents with T1D. According to the findings in the present study, the image-based method appeared to be more accurate among the girls than in the boys; this finding may be due to a small sample size and would need more clarification in future studies. The young participants considered photographing easier and faster compared with the food diary and were more willing to use it in their dietary assessment. However, more detailed instructions and frequent prompts are required to maintain a good compliance with using this method. The image-based method may be a useful dietary assessment tool for adolescents with T1D, but additional research in larger samples is needed.

## Figures and Tables

**Figure 1 nutrients-13-01319-f001:**
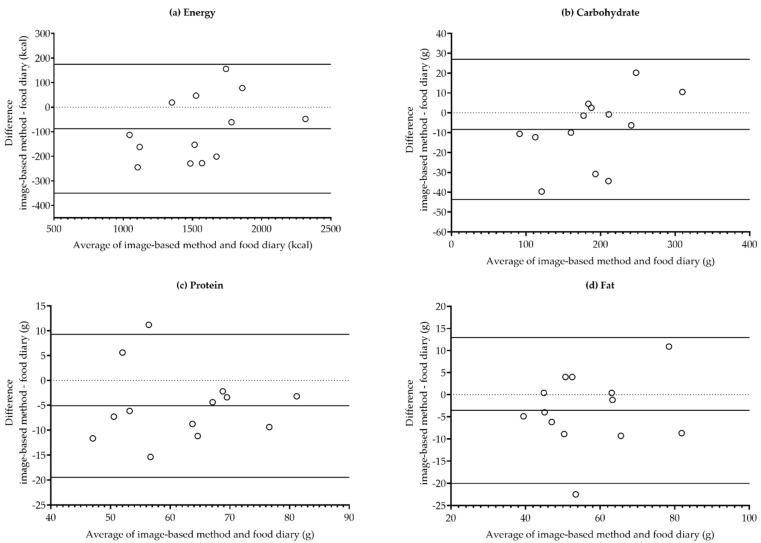
The Bland–Altman limits of agreement plots (mean ± 2 standard deviation units) in the mean daily (**a**) energy; (**b**) carbohydrate; (**c**) protein; and (**d**) fat intake.

**Table 1 nutrients-13-01319-t001:** The characteristics of the study participants.

	Boys (*n* = 4)	Girls (*n* = 9)	Total (*n* = 13)
Age (years), median (IQR)	15.5 (3.3)	15.0 (2.0)	15.0 (2.0)
Height (cm), median (IQR)	164.9 (21.3)	161.7 (10.0)	162.1 (11.5)
Weight (kg), median (IQR)	53.3 (40.8)	56.4 (6.9)	55.4 (7.8)
Diabetes duration (years), median (IQR)	9.1 (8.2)	5.4 (9.9)	8.0 (8.9)
HbA1c (%) ^1^, median (IQR)	8.5 (3.0)	8.0 (1.0)	8.0 (1.0)
Normal weight ^2^, *n* (%)	2 (50)	6 (67)	8 (62)
Daily mobile phone camera use, *n* (%)	1 (25)	5 (56)	6 (46)

HbA1c, hemoglobin A1c; IQR, interquartile range. The data are median (IQR) or *n* (%). ^1^ Self-reported by the participants, latest HbA1c before study entry. ^2^ Defined by weight-for-height [33].

**Table 2 nutrients-13-01319-t002:** The mean energy and nutrient intakes in the original data.

	Boys (*n* = 4)		Girls (*n* = 9)		Total (*n* = 13)	
Energy and Nutrients	Image-Based Method	Food Diary	Image-Based Method	Food Diary	Image-Based Method	Food Diary ^1^
Energy (kcal), median (IQR)	1413 (962)	2351 (852)	1549 (611)	1592 (509)	1455 (586)	1750 (784)
Protein (g), median (IQR)	65.5 (27.8)	84.6 (23.5)	59.3 (16.7)	64.4 (17.8)	59.3 (18.2)	69.3 (29.8)
Carbohydrates (g), median (IQR)	183.3 (159.2)	266.9 (171.0)	185.9 (96.2)	181.4 (87.8)	185.9 (93.6)	225.7 (72.6) **
Fat (g), median (IQR)	44.9 (27.0)	86.5 (39.8)	54.5 (18.9)	50.5 (21.1)	52.7 (19.5)	63.9 (34.3)
SFA (g), mean (SD)	18.1 (5.5)	28.1 (1.9)	20.4 (6.0)	22.2 (6.6)	19.7 (5.8)	24.0 (6.2) *
MUFA (g), median (IQR)	16.8 (10.0)	30.9 (15.2)	19.7 (8.3)	17.3 (9.5)	18.5 (7.8)	24.0 (14.1) *
PUFA (g), median (IQR)	8.4 (5.1)	13.7 (14.9)	8.3 (7.0)	10.0 (6.9)	8.3 (5.6)	11.2 (6.4) **
Sucrose (g), median (IQR)	33.0 (29.9)	26.8 (85.8)	38.7 (35.3)	36.9 (32.0)	38.7 (33.1)	30.6 (32.0) ***
Fiber (g), median (IQR)	8.1 (2.4)	12.4 (8.0)	12.5 (6.0)	13.1 (8.5)	12.0 (6.0)	13.1 (6.9) **
Water-insoluble fiber (g), median (IQR)	5.9 (2.1)	9.3 (6.0)	9.3 (4.6)	9.6 (5.9)	8.7 (4.2)	9.6 (5.3) **
Polysaccharides (g), median (IQR)	2.3 (0.5)	3.4 (2.0)	3.0 (2.9)	3.3 (3.1)	2.6 (1.3)	3.3 (1.8) ***
Vitamin A (µg), mean (SD)	717.1(923.8)	954.7 (954.8)	509.3 (252.1)	579.5 (346.6)	573.0 (515.5)	694.9 (583.5) ***
Vitamin B1 (mg), median (IQR)	1.1 (0.5)	1.5 (0.6)	0.9 (0.3)	1.1 (0.3)	0.9 (0.4)	1.2 (0.4) *
Vitamin B3 (mg), median (IQR)	23.2 (2.6)	34.7 (27.2)	22.6 (5.8)	25.3 (8.1)	22.9 (3.6)	26.2 (11.0)
Folate (µg), median (IQR)	154.8 (117.4)	233.0 (248.0)	148.0 (101.8)	176.1 (107.5)	151.0 (101.8)	177.0 (144.0) ***
Vitamin B6 (mg), median (IQR)	1.6 (0.6)	2.2 (2.6)	1.5 (0.4)	1.6 (0.4)	1.5 (0.5)	1.7 (0.6)
Vitamin B2 (mg), median (IQR)	1.9 (1.4)	2.1 (1.8)	1.5 (0.8)	1.7 (0.7)	1.6 (0.8)	1.8 (0.9) ***
Vitamin B12 (µg), median (IQR)	4.7 (6.6)	4.9 (9.6)	3.4 (1.1)	3.6 (2.5)	3.6 (1.7)	4.0 (2.2) ***
Vitamin C (mg), median (IQR)	47.2 (33.4)	47.5 (148.1)	67.6 (112.8)	74.9 (142.9)	61.4 (44.1)	70.6 (144.0) ***
Vitamin D (µg), mean (SD)	7.3 (2.9)	8.8 (3.3)	7.6 (4.6)	7.8 (4.4)	7.5 (4.1)	8.1 (4.0) ***
Vitamin E (mg), median (IQR)	5.9 (1.5)	9.2 (7.2)	7.8 (5.0)	7.9 (6.1)	6.8 (3.8)	7.9 (5.5) **
Sodium (mg), mean (SD)	2100 (458.9)	3147 (416.4)	2107 (695.0)	2142 (593.3)	2105 (612.1)	2451 (715.0) *
Magnesium (mg), median (IQR)	239.3 (97.3)	302.4 (98.1)	247.1 (70.2)	263.2 (79.2)	247.1 (69.0)	283.9 (67.1) *
Potassium (mg), median (IQR)	2748 (1103)	4065 (1768)	2631 (1080)	3149 (1229)	2713 (1011)	3354 (1119)
Calcium (mg), mean (SD)	988.0 (509.3)	1149 (468.9)	951.9 (310.1)	1051 (329.3)	963 (359.5)	1081 (359.8) **
Iron (mg), mean (SD)	7.3 (1.0)	10.8 (3.3)	8.2 (2.6)	9.0 (2.3)	7.9 (2.2)	9.5 (2.6)
Copper (mg), median (IQR)	0.8 (0.6)	1.1 (1.0)	0.9 (0.4)	1.0 (0.6)	0.9 (0.4)	1.0 (0.5) **
Iodine (µg), mean (SD)	188.6 (65.6)	240.7 (99.0)	177.1 (53.7)	195.0 (53.3)	180.6 (55.0)	209.0 (69.5) **
Salt (mg), median (IQR)	5410 (1802)	7897 (2589)	5613 (3450)	5720 (2669)	5613 (2313)	6470 (3539) *
Phosphorus (mg), median (IQR)	1207 (715.7)	1482 (647.6)	1094 (491.1)	1268 (485.9)	1141 (488.3)	1304 (502.4) **
Selenium (µg), median (IQR)	57.9 (18.9)	65.8 (43.0)	50.2 (12.1)	54.2 (19.1)	51.4 (15.6)	61.9 (18.3) *
Zinc (mg), median (IQR)	9.1 (3.1)	11.0 (5.1)	8.3 (2.5)	9.0 (1.1)	8.7 (2.5)	9.2 (2.3)

IQR, interquartile range; MUFA, monounsaturated fatty acid; PUFA, polyunsaturated fatty acid; SD, standard deviation; SFA, saturated fatty acid. Values are the mean (SD) for the normally distributed and median (IQR) for the non-normally distributed variables. ^1^ Significance of the intraclass correlation coefficient between the methods in the total data: * *p* < 0.05, ** *p* < 0.01, *** *p* < 0.001.

**Table 3 nutrients-13-01319-t003:** The mean energy and nutrient intakes between the image-based method and the food diary in the deducted ^1^ data (*n* = 13).

Energy and Nutrients	Image−Based Method	Food Diary	Difference ^2^	ICC	*p*−Value ^3^
Energy (kcal), mean (SD)	1501 (381)	1589 (330)	−88 (−166, −8) *	0.91 (0.66–0.97)	<0.001
Protein (g), mean (SD)	59.6 (11.0)	64.7 (11.1)	−5.1 (−9.4, −0.8) *	0.72 (0.24–0.91)	<0.001
Carbohydrates (g), mean (SD)	184.0 (64.4)	192.4 (55.9)	−8.4 (−19.0, 2.3)	0.95 (0.84–0.99)	<0.001
Fat (g), mean (SD)	54.8 (14.3)	58.4 (13.0)	−3.5 (−8.5, 1.4)	0.80 (0.48–0.94)	<0.001
SFA (g), mean (SD)	19.7 (5.8)	21.8 (6.0)	−2.1 (−4.4, 0.2)	0.76 (0.36–0.92)	<0.001
MUFA (g), mean (SD)	19.4 (5.9)	20.3 (5.4)	−0.9 (−3.2, 1.3)	0.79 (0.45–0.93)	<0.001
PUFA (g), mean (SD)	9.4 (3.7)	10.0 (3.4)	−0.6 (−1.6, 0.4)	0.88 (0.67–0.96)	<0.001
Sucrose (g), median (IQR)	38.7 (33.1)	33.4 (32.0)	−0.5 (3.1)	0.99 (0.96–1.00)	<0.001
Fiber (g), mean (SD)	12.7 (6.1)	13.5 (5.9)	−0.9 (−1.7, 0.0) *	0.96 (0.86–0.99)	<0.001
Water−insoluble fiber (g), mean (SD)	9.2 (4.2)	9.8 (4.0)	−0.6 (−1.2, 0.0)	0.96 (0.86–0.99)	<0.001
Polysaccharide (g), median (IQR)	2.6 (1.3)	2.7 (1.4)	0.3 (0.5) *	0.95 (0.76–0.99)	<0.001
Vitamin A (µg), mean (SD)	573 (515)	660 (527)	−87 (−175, 2)	0.95 (0.82–0.99)	<0.001
Vitamin B1 (mg), mean (SD)	1.0 (0.2)	1.1 (0.2)	−0.1 (−0.2, 0.0) *	0.76 (0.24–0.93)	<0.001
Vitamin B3 (mg), mean (SD)	22.1 (3.0)	24.4 (4.4)	−2.3 (−4.0, −0.6) *	0.61 (0.06–0.87)	0.002
Folate (µg), mean (SD)	183.5 (83.4)	200.3 (78.0)	16.9 (−33.3, −0.4) *	0.93 (0.73–0.98)	<0.001
Vitamin B6 (mg), mean (SD)	1.5 (0.3)	1.6 (0.3)	−0.1 (−0.2, −0.1) *	0.85 (0.16–0.96)	<0.001
Vitamin B2 (mg), mean (SD)	1.7 (0.6)	1.7 (0.4)	0.0 (−0.2, 0.1)	0.90 (0.70–0.97)	<0.001
Vitamin B12 (µg), median (IQR)	3.6 (1.7)	3.9 (1.9)	0.5 (0.7) *	0.91 (0.71–0.97)	<0.001
Vitamin C (mg), median (IQR)	61.4 (44.1)	70.6 (96.6)	6.0 (26.2)	0.91 (0.75–0.97)	<0.001
Vitamin D (µg), mean (SD)	7.5 (4.1)	7.3 (3.5)	0.2 (−0.5, 0.9)	0.95 (0.85–0.99)	<0.001
Vitamin E (mg), mean (SD)	7.2 (2.4)	8.0 (2.9)	−0.7 (−1.5, 0.0)	0.87 (0.58–0.96)	<0.001
Sodium (mg), mean (SD)	2105 (612)	2196 (540)	−91 (−281, 100)	0.85 (0.59–0.95)	<0.001
Magnesium (mg), mean (SD)	251.7 (69.1)	263.5 (56.9)	−11.8 (−29.6, 6.1)	0.88 (0.67–0.96)	<0.001
Potassium (mg), median (IQR)	2713 (1011)	2818 (1085)	70 (551) *	0.85 (0.46–0.96)	<0.001
Calcium (mg), mean (SD)	963 (360)	986 (321)	−23 (−127, 81)	0.88 (0.66–0.96)	<0.001
Iron (mg), mean (SD)	7.9 (2.2)	8.6 (1.9)	−0.7 (−1.5, 0.1)	0.75 (0.37–0.92)	0.001
Copper (mg), mean (SD)	0.9 (0.3)	1.0 (0.3)	−0.1 (−0.1, 0.0)	0.89 (0.68–0.97)	<0.001
Iodine (µg), mean (SD)	180.6 (55.0)	189.0 (50.4)	−8.4 (−24.9, 8.1)	0.86 (0.63–0.96)	<0.001
Salt (mg), mean (SD)	5328 (1529)	5559 (1370)	−230 (−715, 254)	0.85 (0.59–0.95)	<0.001
Phosphorus (mg), mean (SD)	1181 (306)	1244 (249)	−63 (−137, 11)	0.89 (0.66–0.97)	<0.001
Selenium (µg), mean (SD)	50.0 (8.8)	54.7 (10.0)	−4.7 (−9.0, −0.4) *	0.65 (0.16–0.88)	0.002
Zinc (mg), mean (SD)	8.4 (1.5)	9.0 (1.1)	−0.6 (−1.2, 0.0) *	0.69 (0.23–0.89)	0.001

ICC, intraclass correlation coefficient; IQR, interquartile range; MUFA, monounsaturated fatty acid; PUFA, polyunsaturated fatty acid; SD, standard deviation; SFA, saturated fatty acid. Values are the mean (SD) for the normally distributed and median (IQR) for the non-normally distributed variables. ^1^ Deducted data = the meals that the study participants forgot to photograph were deducted from the original food diary data. ^2^ The mean/median difference for the image-based method—the food diary: * *p* < 0.05. ^3^ Significance of the ICC between the two methods.

## Data Availability

The data presented in this study are available on reasonable request from the corresponding author.

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
