# Peer review of "Agreement between an Image-Based Dietary Assessment Method and a Written Food Diary among Adolescents with Type 1 Diabetes"

_nutrients, 2021, doi:10.3390/nu13041319_

Round 1

Reviewer 1 Report

General feedback

This was a feasibility study which explored the agreement between an image-based food record with an written food diary for energy and nutrient intake amongst a group of adolescents with type 1 diabetes mellitus. Given the participant burden of keeping a food record as well as the potential for inaccuracies in keeping a written diary by adolescents, this study addressed an important research question in the area of dietary methodology.

Overall, the paper needs English language editing to improve the scientific writing style by writing more succinctly as well as correct grammatical errors that is evident throughout the manuscript. Some of which are commented on in the manuscript.

Major revisions are recommended for the methods section, currently there are gaps in the research methodology and sample size calculation which need to be addressed. Lack of fiducial marker or use of standardised plate sizes is a major limitation in the study design, as well as the lack of consistency in the record keeping method of portion sizes in the food diary.

Specific feedback

Methods

Clarification of study design and sample size calculation needs to be included.

It is not clear why informed consent from parents or guardians would not be applicable of minors. Please clarify.

Further clarification is needed on how unclear photographs were managed.

Further clarification and justification of how portion sizes were captured by participants when completing the food diaries is required.

Further clarification is required on additional detail provided with photographs to researchers regarding food quality and fat content.

Development of the survey on participant experience needs to be included and broad description of types of questions should be included rather than details of the questions.

Statistical analysis

It is not clear why analyses of data set containing incomplete records are included, it is expected to be different when compared to a more complete data set. It is recommended to remove these results from the manuscript.

Results

Presentation of results in text need to be written using a plus-minus sign when reference is made to standard deviation.

Results of Table 1 is not described in the methods section, this needs to be included.

Reference is made to Interquartile ranges in tables, however, it seems that standard deviations are reported.

Table 1 refers to weight-and-height standards, include which standards were used e.g. CDC and make reference to body mass index.

Reference is made to ‘moderate to excellent’ energy intake estimations in Table 3, it is not clear how this was classified. Include description in statistical analysis section.

Table 3: Only include one unit of energy calculation. Include all p-values as a new column. Reference is made to a number of p-values in the footnote, however only one symbol is used in the table.

Fig 1: Further in-text description is required to the findings presented in Fig 1a to d.

Section 3.2: Include both frequencies in percentages in text and indicated percentage participants selecting reasons for easiness of use of the image-based method.

Discussion

Lines 213-222: Revise this section to highlight key findings without repeating results.

Lines 223 to 224 contradicts what is said in lines 214-215.

Lines 240-241: Provide further detail and context of previous studies. What might explain similarities?

Lines 241-244: It is not clear why reasons for misreporting of protein intake is discussed if underestimation of protein intake in the current study was negligible.

Lines 249-250: Provide further discussion on potential reasons for comparable findings to those of others.

Line 251-252: What is the relevance to include a study on adults here?

Author Response

Responses to the Reviewer 1

Methods

Clarification of study design and sample size calculation needs to be included.

Response: We thank the reviewer for these suggestions and have now clarified in the text that this comparative study included the image-based method as a test method and the food diary as a standard method (lines 82-83). An a priori sample size calculation was not performed in this study. We recruited volunteer patients from the Pediatric Diabetes Outpatient Clinic at Oulu University Hospital during June to October 2015 and June 2016. Altogether 16 adolescents gave their consent to the study, and 13 of them were included in the analyses. We agree with the reviewer that a larger sample size would naturally have been more convincing.

It is not clear why informed consent from parents or guardians would not be applicable of minors. Please clarify.

Response: According to the Finnish legislation, if the minor has reached the age of 15 and can understand the research, it shall be sufficient for the minor to give his/her informed consent in writing. Therefore, a written informed consent was not asked for parents or guardians if the minor was 15–18 years old and capable of giving his/her informed consent. Regarding 13–14-year-old participants, both the minors and their participants or guardians gave their informed consents. The more detailed information has now been provided in the lines 80-81.

Further clarification is needed on how unclear photographs were managed.

Response: If the photo was unclear according to the judgement of the participant, he/she was advised to take a new one (lines 88-89). All the photos were asked to be sent to the researcher within one day. If the researcher noticed an unclear photo, the participant was asked for additional information (line 99-101). The previous sentence states that the subjects were asked to make sure all ingredients clearly seen in the photo (lines 85-87).

Further clarification and justification of how portion sizes were captured by participants when completing the food diaries is required.

Response: We agree with the reviewer that the use of one standardised approach is recommended and using uncalibrated scales can result in inaccuracies. However, it could be difficult for adolescents to provide all portion sizes as weighed values in free-living conditions. Thus, we decided to allow different approaches to make portion size estimation as detailed as possible and use of uncalibrated scales was also allowed.

Food weights were calculated according to the portion sizes using the Finnish food composition database Fineli. If the pieces meant the number of food items (e.g., two apples), standard weights of them were used in the analyses. If the pieces meant piece of food (e.g., piece of apple pie), its portion size was instructed to be provided as centimetres. If the exact size of the piece was not provided, standard portion sizes were used. Drawings of portion sizes were not used in this study. Further information has now been provided in the lines 93-94.

Further clarification is required on additional detail provided with photographs to researchers regarding food quality and fat content.

Response: Food quality and fat content were applicable to processed or cooked food. Food quality meant for example fat-free or semi-skimmed milk, sugared or diet soft drink and wheat or oat bread. Brands of packaged food were also provided in additional information. The more detailed description has now been added in the lines 97-99.

Development of the survey on participant experience needs to be included and broad description of types of questions should be included rather than details of the questions.

Response: Details of questions have now been replaced with a broad description of questions (lines 111-132). The group of experienced dietitians and physicians developed the questionnaire for this study, and our aim was to pilot the questionnaire practically in this study.

Statistical analysis

It is not clear why analyses of data set containing incomplete records are included, it is expected to be different when compared to a more complete data set. It is recommended to remove these results from the manuscript.

Response: We thank the reviewer for this suggestion. However, in addition to the agreement between the two methods, we aimed to evaluate feasibility of the image-based method in this study. Hence, original data including underestimated energy intake are also provided.

Results

Presentation of results in text need to be written using a plus-minus sign when reference is made to standard deviation.

Response: The brackets have now been replaced with plus-minus signs (lines 189, 212, 216, 229, and 230).

Results of Table 1 is not described in the methods section, this needs to be included.

Response: We thank the reviewer for this important remark. The background questionnaire and height and weight measurements have now been described in the methods section (lines 104-108).

Reference is made to Interquartile ranges in tables, however, it seems that standard deviations are reported.

Response: The interquartile ranges in the text and the tables have been represented as one number (Q3–Q1).

Table 1 refers to weight-and-height standards, include which standards were used e.g. CDC and make reference to body mass index.

Response: In Finland, we have national population-based growth references for children and adolescents and normal weight for adolescents was defined according to these standards. A reference to more detailed description of the Finnish growth references is given in the line 188 and the Table 1.

Reference is made to ‘moderate to excellent’ energy intake estimations in Table 3, it is not clear how this was classified. Include description in statistical analysis section.

Response: Many thanks for this comment. The description with a reference has now been added in the statistical analyses section (lines 177-179).

Table 3: Only include one unit of energy calculation. Include all p-values as a new column. Reference is made to a number of p-values in the footnote, however only one symbol is used in the table.

Response: Energy intake has now been reported only using kilocalories as it has been reported in the text. The exact p-values have been represented in the Table 3 as a new column and the footnote concerning significance of the ICCs has been removed. Useless symbols for p-values have also been removed.

Fig 1: Further in-text description is required to the findings presented in Fig 1a to d.

Response: The findings shown in Figure 1 have now been described in the lines 230-234.

Section 3.2: Include both frequencies in percentages in text and indicated percentage participants selecting reasons for easiness of use of the image-based method.

Response: The frequencies have now been added in percentages to the section 3.2.

Discussion

Lines 213-222: Revise this section to highlight key findings without repeating results.

Response: The first chapter of the discussion has now been edited to better highlight key findings of the study (lines 258-265)

Lines 223 to 224 contradicts what is said in lines 214-215.

Response: We thank the reviewer for this viewpoint. Agreement between the image-based method and food diary was moderate to excellent in energy intake estimation. However, difference in energy intake between the methods differed significantly from zero, and therefore, the image-based method has been written to underestimate energy intake.

Lines 240-241: Provide further detail and context of previous studies. What might explain similarities?

Response: Additional information on the previous studies has now been provided (lines 289-292). Comparable findings may be due to quite similar studies among children and adolescents. Lower correlations for protein intake estimates may relate to for example adolescents’ difficulty to estimate protein-rich foods’ portion sizes.

Lines 241-244: It is not clear why reasons for misreporting of protein intake is discussed if underestimation of protein intake in the current study was negligible.

Response: We thank the reviewer for this good comment. The discussion of protein intake underestimation has now been removed (lines 292-297).

Lines 249-250: Provide further discussion on potential reasons for comparable findings to those of others.

Response: Comparable findings may be due to quite similar studies, for example both the two previous studies have also compared image-based methods with written food diaries and the participants have been children and adolescents. Underreporting of energy intake has previously been observed in most of image-based dietary assessment studies.

Line 251-252: What is the relevance to include a study on adults here?

Response: Most of image-based dietary assessment studies have included adults and only a few studies have compared image-based methods with written food records among adolescents. Therefore, we decided to compare our findings also with a previous review among adults (lines 303-304).

Other changes: 

English language was edited according to the reviewer’s and the English editors’ feedback.

Reviewer 2 Report

Thank you for the opportunity to review this interesting report that evaluates the accuracy of a photo based assessment versus the current four day food diary for dietary assessment. This is a very topical area as increasingly new technologies to support food estimations and to assist carbohydrate counting accuracy in T1D are being developed. The paper is well written, however there are two main limitations that require further discussion by the authors.

  1. The method of photo assessment used in this report seems quite laborious for the researcher and relies on picture booklets of portion sizes to assist researchers to quantify amounts. To assess the generalisability of the tool and findings can the authors please include the approximate time spent by researchers on portion estimation, the number and content of messages exchanged with participants to gain additional information and if portion sizes can be estimated without the assistance of the booklets of photographs of commonly eaten foods? How were food quantities estimated from the photographs if foods were not in the books? Can the authors comment on inclusion of standard tool such as a teaspoon in the food photos to assist portion size estimations as in other studies?  
  2. Missing food photos result in highly significant under-estimation of the energy and nutrient content. This is not an issue in the food diary method. Do the authors think this supports the need for additional food record keeping with the photo assessment method?

The findings therefore require softening to reflect the tendency of adolescents to neglect to take photos of all foods which significantly impacts accuracy of estimation.

Additional comments follow:

  1. Please explain sample size calculation (n=13, n=264 meals). Based on small numbers it seems gender differences should not be explored.
  2. Was there any association between meal type (breakfast, lunch, dinner and snack), food type (cereal, fruit and packaged goods), meal size (large carbs, small carb quantities) and accuracy of photographic method?
  3. The micronutrient analysis for the image based method (Table 2) does not add as the energy intake was highly significantly under-reported and thus impacts all other nutrients.
  4. The introduction is very broad and would benefit from a review of the literature of methods of carbohydrate assessment using imagery specifically in diabetes management
  5. Ethics approval was gained in 2015. Has the image method been further developed or modified or gained commercial approval? Some discussion of future directions would be helpful.
  6. Table 1 requires mean HbA1c for participants
  7. Figure 1- the number of Figures requires rationalisation. This would benefit from more of a focus on nutrients impacting diabetes management- carbohydrate.

Author Response

Responses to the Reviewer 2

Thank you for the opportunity to review this interesting report that evaluates the accuracy of a photo based assessment versus the current four day food diary for dietary assessment. This is a very topical area as increasingly new technologies to support food estimations and to assist carbohydrate counting accuracy in T1D are being developed. The paper is well written, however there are two main limitations that require further discussion by the authors.

1. The method of photo assessment used in this report seems quite laborious for the researcher and relies on picture booklets of portion sizes to assist researchers to quantify amounts. To assess the generalisability of the tool and findings can the authors please include the approximate time spent by researchers on portion estimation, the number and content of messages exchanged with participants to gain additional information and if portion sizes can be estimated without the assistance of the booklets of photographs of commonly eaten foods? How were food quantities estimated from the photographs if foods were not in the books? Can the authors comment on inclusion of standard tool such as a teaspoon in the food photos to assist portion size estimations as in other studies?  

Response: The time spent on portion size estimation and number of messages exchanged with participants were not recorded in this study. We totally agree with the reviewer that it would have been important and should be done in future studies. The messages for participants focused on missing further details, such as fat content of milk or margarine. According to our experiences, portion sizes can also be estimated without the assistance of the booklets but then a fiducial marker should be used to increase accuracy of portion size estimation. If foods in the photos were not in the booklets, portion sizes were visually estimated compared with dishes and packages. Brands of foods and drinks and package labels seen in the photographs were also used. We edited the discussion according to these good questions (lines 328-333).

2. Missing food photos result in highly significant under-estimation of the energy and nutrient content. This is not an issue in the food diary method. Do the authors think this supports the need for additional food record keeping with the photo assessment method?

Response: We totally agree with the reviewer that missing photos significantly weakened feasibility and validity of the image-based method. However, keeping food record simultaneously with the image-based method would notably increase participants’ burden. For future studies, we suggest that it would be useful to develop a dietary intake mobile app for research purposes. The app would not only use a mobile phone camera but also record further details, give prompts, provide a backup method, and include a feature to contact study participants to reduce missing data (lines 321-323 and 343-345).

The findings therefore require softening to reflect the tendency of adolescents to neglect to take photos of all foods which significantly impacts accuracy of estimation.

Response: The abstract (lines 25-28), first chapter of discussion (lines 258-269), and the conclusions (lines 361-367) have now been edited to soften the main findings and to highlight adolescents’ tendency to neglect to take photos.

Additional comments follow:

1. Please explain sample size calculation (n=13, n=264 meals). Based on small numbers it seems gender differences should not be explored.

Response: An a priori sample size calculation was not performed in this study. We recruited volunteer patients from the Pediatric Diabetes Outpatient Clinic at Oulu University Hospital during June to October 2015 and June 2016. Altogether 16 adolescents gave their consent to the study, and 13 of them were included in the analyses. We agree with the reviewer that a larger sample size would naturally have been more convincing. Although the sample size was small, the results in the tables are also represented separately for boys and girls, because the image-based method seemed to be more accurate among girls than boys.

2. Was there any association between meal type (breakfast, lunch, dinner and snack), food type (cereal, fruit and packaged goods), meal size (large carbs, small carb quantities) and accuracy of photographic method?

Response: We thank the reviewer for these interesting questions. Associations between meal type, food type, meal size, and accuracy of the image-based method were not analyzed in this study, but snacks and some types of food (e.g., spreads) seemed to be most often unreported.

3. The micronutrient analysis for the image based method (Table 2) does not add as the energy intake was highly significantly under-reported and thus impacts all other nutrients.

Response: In addition to agreement between the two methods, we aimed to evaluate feasibility of the image-based method in this study. Hence, original data including underestimated energy intake are also provided.

4. The introduction is very broad and would benefit from a review of the literature of methods of carbohydrate assessment using imagery specifically in diabetes management

Response: We thank the reviewer for this suggestion. However, only a few studies have examined image-based methods in carbohydrate estimation among people with diabetes and these studies have included adults. We added the mention of novel techniques for carbohydrate intake estimation in diabetes using photographs or voice (lines 52-54). To our knowledge, this is the first study concerning the image-based method in dietary intake estimation among adolescents with diabetes. Thus, the introduction focuses on previous dietary intake estimation studies among children and adolescents in general.

5. Ethics approval was gained in 2015. Has the image method been further developed or modified or gained commercial approval? Some discussion of future directions would be helpful.

Response: We have not further developed the image-based method in our studies, but some suggestions for future development are provided in the discussion (lines 320-323, 332-333, and 343-345).

6. Table 1 requires mean HbA1c for participants

Response: The median HbA1c values have now been provided in the Table 1.

7. Figure 1- the number of Figures requires rationalisation. This would benefit from more of a focus on nutrients impacting diabetes management- carbohydrate.

Response: We thank the reviewer for this viewpoint. However, the image-based method should be evaluated not only for diabetes management but also to estimate diet quality and adherence to healthy dietary habits among adolescents with type 1 dietary. Hence, the Bland–Altman plots for energy, protein and fat intakes are also provided in Figure 1.

Round 2

Reviewer 1 Report

Thank you for your revisions. Unfortunately there are still a number of grammatical errors, unclarity in the methods section and reporting errors in the results section. See some comments attached in the revised manuscript for your consideration.

Author Response

Thank you for your revisions. Unfortunately there are still a number of grammatical errors, unclarity in the methods section and reporting errors in the results section. See some comments attached in the revised manuscript for your consideration

Response: Many thanks for your additional comments. Our manuscript has undergone English language editing by MDPI. Two editors have checked the text for correct use of grammar and common technical terms, and edited to a level suitable for reporting research in a scholarly journal. The certificate of English editing can be found at https://odlfi-my.sharepoint.com/:b:/g/personal/laura_heikkila_odl_fi/EQ8__9zGm1ZMmnQsKLhnyd4B0u4ckhR6LyByVpy7XtOGcA?e=wscQvJ.

Detailed feedback in the revised manuscript

Lines 88-89: When was done and by whom? Did the researcher ask the participant to retake the photo or were the participant instructed what a clear picture looks like and judged for themselves before sending?

Response: The participants were instructed to ensure that all the foods and drinks are clearly seen in the photos but detailed criteria for a clear photo were not provided. The participants judged for themselves the photos after taking. The text has been edited to better describe when the photos were judged and by whom (lines 88-89).

Line 94: what about size? small, medium large? how was portion size estimated?

Response: The participants themselves estimated portion sizes as accurate as possible according the instructions. The description of portion sizes was also instructed (for example a small, medium or large banana). The sentence has been edited and some additional examples have been included (lines 94-96).

Line 94: unit correct?

Response: The example of unit (dL) is correct and accepted by MDPI journals. The participants reported for example portion sizes of drinks as deciliters in our study (line 95).

Lines 102-102: How was these records then checked for completeness and accuracy?

Response: Thank you for this comment. The completeness of the food diaries were checked after the four-day study period. One participant did not complete the food diary and therefore, she was excluded from the study. We agree that the accuracy of the food diaries should have been checked after the study period. The inspections of the food diaries have been discussed as limitations (lines 327-329 and 358-360).

Line 106: provide an estimate time of last acceptable measurement.

Response: The participants were recruited from the diabetes nurse visits at the Oulu University Hospital. At the visits, all the participants’ hemoglobin A1c value were measured. Most of the participants replied to the background questionnaire within two weeks of their visits and one participant approximately 2.5 months after his visit. Hence, the longest acceptable time between measurement and questionnaire was three months. An estimate time of last acceptable measurement has now been provided in the lines 108-109.

Line 112: rewrite: (5= description and 1=description)

Response: Thank you for this suggestion. The sentence has now been revised (line 115).

Line 129-130: keep this question in.

Response: The question has now been included in the text (lines 132-133). We also included the question focusing on the factors restricting the use of the image-based method (lines 127-128).

Lines 131-132: strikethrough

Response: The answer options for the question have been removed (lines 134-135).

Line 139: It would have been useful to provide participants with these booklets to help them estimate portion sizes when they kept their food diaries for example when scales were not available.

Response: We agree that food picture booklets would also be useful for study participants in portion size estimation.

Table 2: IQR indicated here but not included in the results.

Response: Table 2 shows median and IQR (or mean and SD for the continuous variables) values for the energy and nutrient intakes in the original data. The IQR values are represented as one number for each median, and the number describes length of the interquartile range (range between the first and the third quartile). The IQR values in the Table 2 are not also reported in the text of results section in order to avoid double reporting. In the results section text, IQR values are provided for difference variables (lines 202-205).

Table 3: Indicate in footnote which p-value this points to? ICC or Difference?

Response: Thank you for this important remark. The footnote for p-value has now been added in the Table 3.

Line 230: limits of agreement and confidnce intervals should be reported

Response: The limits of agreement and confidence intervals have now been reported in the lines 234-235 and 238-239.